# Influence of Butorphanol, Buprenorphine and Levomethadone on Sedation Quality and Postoperative Analgesia in Horses Undergoing Cheek Tooth Extraction

**DOI:** 10.3390/vetsci9040174

**Published:** 2022-04-06

**Authors:** Daphna Emanuel, Sabine B. R. Kästner, Julien Delarocque, Anne J. Grob, Astrid Bienert-Zeit

**Affiliations:** 1Clinic for Horses, University of Veterinary Medicine Hannover, Foundation, Buenteweg 9, 30559 Hannover, Germany; julien.delarocque@tiho-hannover.de (J.D.); anne.julia.grob@tiho-hannover.de (A.J.G.); astrid.bienert@tiho-hannover.de (A.B.-Z.); 2Clinic for Small Animals, University of Veterinary Medicine Hannover, Foundation, Buenteweg 9, 30559 Hannover, Germany; sabine.kaestner@tiho-hannover.de

**Keywords:** analgesia, dental pain, equine, opioids, pain management, pain scoring

## Abstract

The aim of this prospective clinical trial was to compare the influence of butorphanol, buprenorphine and levomethadone on sedation quality and postoperative analgesia in horses undergoing cheek tooth extraction. Fifty horses were assigned to three groups prior to oral cheek tooth extraction. Horses were treated with acepromazine, followed by a detomidine bolus, one of the three opioids and both a nerve block and gingival anaesthesia. During the surgery, sedation was maintained with a detomidine constant rate infusion. After surgery, the quality of sedation, surgical conditions and severity of the extraction were assessed with a numerical rating scale. To evaluate differences in the quality of analgesia between the three treatments, postoperative pain was estimated with the Equine Utrecht University Scale for Facial Assessment of Pain. Additionally, several parameters that are associated with dental pain were added to this validated pain score, and blood samples were taken to measure serum cortisol. Our analysis showed lower pain scores and a greater analgesic effect with levomethadone and buprenorphine compared with butorphanol, with increased locomotor activity induced by buprenorphine. While cortisol values demonstrated higher response in horses treated with levomethadone and buprenorphine compared to butorphanol, these values could be biased by unrelated stressors.

## 1. Introduction

Standing sedation in horses is becoming increasingly popular for surgeries such as cheek tooth extractions to avoid the risks associated with general anaesthesia and injuries due to recovery. It requires deep sedation while maintaining patient’s ability to stand. To achieve this, a reliable sedation protocol with sufficient multimodal analgesia is required. Alpha (α)2-agonists in combination with opioids are the most commonly used drugs for sedation. They lead to dose-dependent sedation but also provoke ataxia and cardiovascular depression [1]. To alleviate cardiovascular effects of α2-agonists, constant rate infusions (CRI) are administered to provide steady plasma concentrations and a constant level of sedation [2,3,4]. Detomidine doses of 20 µg kg^−1^ h^−1^ are required to maintain the sedation of horses undergoing cheek tooth extractions [5,6].

Inclusion of opioids in the sedation protocol can provide multimodal analgesia and enhance sedation [4,7,8]. In particular, the opioid butorphanol (BUT) is widely used in equine medicine in Germany because it is not subject to the Narcotics Act (Betäubungs-mittelgesetz). It is a synthetic opioid receptor agonist–antagonist and conveys a short (60–90 min) and reliable visceral analgesia but may not be the ideal opioid of choice for somatic pain, which is the type of pain caused by dental disease and treatments [9,10,11,12]. Somatic pain can be divided into superficial and deep somatic pain. Superficial somatic pain has a stabbing, burning character. An example in the orofacial region can be mucogingival pain caused by burns, gingivitis, herpes infections or chronic recurrent aphthae pain character. Deep somatic pain in the orofacial region includes pulpal pain and temporomandibular joint pain, as well as pain of a desmodontal origin—i.e., originating from the periodontal ligament. The character of deep somatic pain is described as “dull” [13,14]. 

Buprenorphine (BUP) is an additional opioid approved for horses in Germany. As a partial-µ-receptor agonist, it shows a long-lasting analgesic effect up to 7 to 9 h after intravenous (IV) administration [15,16]. It was demonstrated to provide improved sedation quality in horses undergoing dental extractions and a reliable analgesia in the treatment of somatic pain [6,17]. Nevertheless, some side effects that can influence the interpretation of the intensity of pain—such as central excitement with manic behaviour, tense facial features and increased locomotor activity—have been reported [6,18]. 

Levomethadone (LEV) is a levorotatory enantiomer of the racemate methadone, which is licensed in combination with an anticholinergic agent fenpipramide for use in horses and dogs [19]. In addition to its effect as a full µ-agonist, it also acts as an NMDA (N-methyl-D-aspartate) receptor antagonist. NMDA receptor antagonists are helpful in preventing the development and maintenance of central sensitization and in significantly reducing the hyperalgesia caused by the wind-up phenomenon—processes that play an important role in the development of chronic pain stages [20,21,22].

Acepromazine is a tranquilizer of the phenothiazine class with a duration of action of about 2 h [23,24]. Phenothiazines have depressant effects in the chemoreceptor trigger zone and in the hypothalamus which may play a role in inhibiting opioid-induced excitation, thereby reducing some of the side effects reported for BUP [25].

Due to their duration of action and specific receptor affinities, we hypothesized that LEV and BUP are better suited analgesics for the treatment of dental pain and during tooth extractions compared with commonly used BUT. Therefore, this study aimed to compare intraoperative sedation quality and postoperative analgesia when using the opioids LEV or BUP as opposed to the commonly used BUT. 

Pain assessment systems that have been developed for the detection and grading of pain are tools used to assess postoperative analgesia. Localization and type of pain cause different pain signs. Therefore, a single assessment system is not suitable for pain assessment of visceral vs. somatic, acute vs. chronic, and nociceptive vs. inflammatory vs. neuropathic pain. The Equine Utrecht University Scale for Facial Assessment of Pain (EQUUS-FAP) proved to be helpful in evaluating and grading acute and postoperative pain in the head region [26]. However, dental pain in horses can be subtle and is often detected during late stages of the pathology. As a result, dental diseases are often diagnosed as an incidental finding during a routine dental examination. This indicates that certain behavioural patterns of horses, which could be associated with dental pain, are not recognized [27]. 

So far, there is no validated pain assessment system specifically designed for pain of dental origin in horses. Typical behavioural changes that are related to tooth- or oral cavity-associated pain such as altered or suspended feed intake, as well as pain on palpation of the affected region of the head, are not recorded in any of the existing pain scores.

In addition to pain scores, increase in blood serum levels of various hormones, such as cortisol, can be used as a stress indicator. As pain is one of the possible causes for stress, the estimation of serum cortisol concentrations can be a helpful tool for the evaluation of pain states [28,29]. 

## 2. Materials and Methods

### 2.1. Study Design

The study was designed as a prospective clinical trial and was approved by the LAVES (Lower Saxony State Office for Consumer Protection and Food Safety) (33.19-42502-05-1IA4 41).

### 2.2. Animals and Group Selection

For the purpose of this study, 50 horses that were referred to the Clinic for Horses of the University of Veterinary Medicine Hannover with dental diseases, and the indication for extraction of one to three cheek teeth was selected as the study population. Only horses with oral extraction were included. To ensure a paired randomized and blinded trial, an independent person (A.J.G.) randomly assigned patients to one of three treatment groups: BUT, LEV or BUP. To ensure that the groups were populated with comparable patients, partner matching was performed. Specifically, a horse that was comparable with another horse already assigned to one group, in terms of age, breed, and tooth to be extracted, was assigned to one of the two other groups.

### 2.3. Preparation, Premedication and Treatment

All surgeries were performed in the morning. Three and a half hours before the procedure, an intravenous catheter (Intraflon^®^ 2 fluoropolymer (PTFE), 13G-L.80 mm, VYGON GmbH & Co.KG, Aachen, Germany) was inserted into one of the jugular veins following aseptic preparation of the skin and subcutaneous infiltration of 2 mL lidocaine (Lidocainhydrochlorid 2%, bela-pharm GmbH & Co. KG, Vechta, Germany). One hour preoperatively, all horses received 0.6 mg kg^−1^ meloxicam IV (Melosolute^®^, 20 mg/mL, CP-Pharma Handelsgesellschaft mbH, Burgdorf, Germany), followed by 50 µg kg^−1^ acepromazine IM (Tranquisol^®^ 10 mg/mL, CP-Pharma Handelsgesellschaft mbH, Burgdorf, Germany) which was administered once 30 min before the detomidine-mediated sedation and a second time two hours after the first application. Additional meloxicam as anti-inflammatory and for pain management (Melosus^®^ 15 mg kg^−1^, CP-Pharma Handelsgesellschaft mbH, Burgdorf, Germany) was administered (0.6 mg kg^−1^ PO) for four days postoperatively.

### 2.4. Sedation Protocol

All horses were sedated with detomidine (Cepesedan^®^ 10 mg/mL, CP-Pharma Handelsgesellschaft mbH, Burgdorf, Germany) (15 µg kg^−1^ IV). This was followed by a 10 min gap before either BUT (Butorgesic^®^ 10 mg/mL, CP-Pharma Handelsgesellschaft mbH, Burgdorf, Germany) (100 µg kg^−1^ IV), LEV (100 µg kg^−1^ IV) (L-Polamivet^®^ Levomethadonhydrochlorid 2.5 mg/mL Fenpipramidhydrochlorid 0.125 mg/mL, Intervet Deutschland GmbH, Unterschleißheim, Germany) or BUP (Bupresol^®^ 0.3 mg/mL, CP-Pharma Handelsgesellschaft mbH, Burgdorf, Germany) (5 µg kg^−1^ IV) was administered. The opioid bolus was masked by dilution with saline to a standard volume of 50 mL. Following an additional 10 min gap, a single bolus of diazepam (Ziapam^®^ 5 mg/mL, Ecuphar GmbH, Greifswald, Germany) (0.01 mg kg^−1^ IV) was administered, and a detomidine CRI was initiated (20 µg kg^−1^ h^−1^) [6].

The depth of sedation was assessed using a sedation score in ten-minute intervals *(*Appendix A) [6]. In cases of insufficient sedation—i.e., defence movements with a single score value of >3—a detomidine bolus of 3 µg kg^−1^ was administered. This was followed by an increase in the CRI by 10 µg kg^−1^ h^−1^. In cases of tongue movements >3, a new bolus of diazepam (0.01 mg kg^−1^ IV) was administered. If the horse showed an ataxia score of >3, the CRI was reduced by 10 µg kg^−1^ h^−1^. Heart rate, respiratory rate and rectal temperature were determined at 10 min intervals. 

Maxillary and/or mandibular nerve block was performed with 2 or 3 mL/100 kg mepivacaine hydrochloride (Mepidor^®^ 20 mg/mL, Richter Pharma AG, Wels, Austria). In addition, the gingiva was locally infiltrated with 20 mL lidocaine. Oral cheek teeth extraction was performed as described by Tremaine (2004) [30] by the same experienced surgeon (A.B.Z.).

### 2.5. Assessment of Sedation Quality, Surgical Conditions and Severity of Extraction

At the end of each surgery, the quality of sedation and surgical conditions and the severity of the extraction procedure was assessed by the surgeon using a numerical rating scale (NRS), from 1 = excellent quality of sedation and surgical conditions, easy extraction to 10 = surgery not feasible, extraction was unsuccessful, extremely elaborate extraction. 

### 2.6. Pre- and Postoperative Measurements

#### 2.6.1. Pain Scores

Equine Utrecht University Scale for Facial Assessment of Pain (EQUUS-FAP) with a range of 0–18 was used for pain scoring [31]. In a second step, the EQUUS-FAP was supplemented with further parameters that might be relevant to detect dental pain (Table 1). Since the supplemented part (SUPP) of the pain score is not validated, it was considered separately in the statistical analysis. To obtain a baseline, the first scoring was performed prior to surgery with postoperative assessment carried out after 3, 6 and 24 h.

#### 2.6.2. Blood Samples

For investigation of pre-, peri- and postoperative stress levels and a possible influence of the chosen opioid, solid phase, competitive, chemiluminescent enzyme immunoassay (LKCO1, ImmuliteTM 1000, Siemens Diagnostics, Malvern, PA, USA) was performed to assess the serum cortisol concentrations. For this purpose, blood was taken at 3 hourly intervals preoperatively, as well as 5 min after initiation of the surgery. Additional samples were taken at 15 min intervals between 30 and 120 min., as well as 150 and 180 min. and 6 h postoperatively. All samples were drawn into VACUETTE^®^ Serum Clot Activator Tubes, Greiner Bio-One GmbH, Kremsmuenster, Austria). 

#### 2.6.3. Locomotor Activity

To compare the effect of opioids on locomotor activity, steps were counted in all patients using pedometers (Pedometer Ex Distance, Silva, Bromma, Sweden) attached to one front limb and diagonal hind limb. Number of steps was counted preoperatively over a period of twelve hours to obtain a baseline value and postoperatively over twelve hours.

### 2.7. Data Analysis

Statistical analysis was performed using R 4.1.0 [32]. Statistical significance was accepted at 0.05.

An a priori power analysis was conducted based on previously measured serum cortisol concentrations during tooth extraction with an effect size (f) of 0.77 [6]. At a statistical significance level of 0.05 for 3 groups and 7 measurements, assuming sphericity, this indicated that a total of 90 horses would be necessary. Since additional measurements were performed at time points suspected to yield stronger group differences, an interim analysis was performed after inclusion of 50 horses in the study. The latter was able to detect statistically significant group differences for serum cortisol, so that a posteriori power was deemed sufficient.

The Kruskal–Wallis test was used for count data and parameters on the ordinal scale (number of detomidine boli, number of defensive movements, ataxia score, median and maximal pain scores, number steps). If applicable, Dunn’s test was used for post hoc comparisons. For the quality of sedation and surgical conditions, where the severity of surgery was an additional predictor besides the used opioid, the Scheirer–Ray–Hare test was used.

The mean rate of the detomidine CRI, as well as the total detomidine dose, was compared across groups using a one-way ANOVA, whereas the evolution of the heart and respiratory rates were evaluated over time with a two-way repeated measures ANOVA. The corresponding assumptions of homoskedasticity or sphericity were assessed using a Levene or Mauchly test, respectively. In the latter case, the provided *p* values were corrected using the Greenhouse–Geisser procedure. Model residuals were inspected visually for deviations from normality. The effect size was described as epsilon squared (ε^2^) when a Kruskal–Wallis test was used, and as generalised eta squared (η^2^_G_) when an ANOVA was performed.

The time course of the pain scores was analysed using a generalised additive model on the ordinal scale using the mgcv R package [33,34]. One model was fitted for each score (FAP, SUPP and FAPplusSUPP) with time modelled as a thin-plate spline and horse modelled as a random effect. It was decided by comparison of model AIC whether an opioid should be modelled as a main effect only or whether an interaction with time should be included. Model fit was ensured visually and using the recommended built-in diagnostics of the mgcv package.

A similar approach was taken for serum cortisol, which was modelled using a generalised additive model of the gamma family with an inverse link function. The *p*-values for the parametric effects (opioid) were computed using Wald tests, as described in the package documentation.

## 3. Results

### 3.1. Animals

The distribution within the individual groups is listed in Table 2.

### 3.2. Sedation and Ataxia

There were neither significant differences in the number of additive detomidine boli between the three opioids (H(2) = 1.21, *p* = 0.55, ε^2^ = 0.02) nor in the mean dripping rate of the CRI (F(2, 47) = 2.02, *p* = 0.144; η^2^ = 0.08) between the groups. Based on the applied sedation score, no significant difference could be estimated in the defensive movements by the choice of opioid (H(3) = 0.123, *p* = 0.94, ε^2^ = 0.002). The choice of the opioid did not lead to further significant changes in the heart (F(2, 47) = 0.08, *p* = 0.92, η^2^_G_ = 0.002) or breathing (F(2, 47) = 0.90, *p* = 0.41, η^2^_G_ = 0.018) rate. Figure 1 shows the differences in the grade of ataxia between the three groups (H(2) = 9.50, *p* = 0.009, ε^2^ = 0.19). While the median scores of the stability show that the patients of the group LEV were significantly less ataxic compared to the group BUT (*p* = 0.006), there was neither a significant difference between LEV and BUP (*p* = 0.18) nor between BUP and BUT (*p* = 0.18).

### 3.3. Assessment of Sedation Quality, Surgical Conditions and Severity of Extraction

The surgical conditions, as well as the severity of the extraction were scored with a median of 3 for all groups. The opioid (H(2) = 1.638, *p* = 0.44, ε^2^ = 0.03) and the severity of extraction (H(7) = 9.449, *p* = 0.22, ε^2^ = 0.19) did not significantly influence the surgical conditions. Furthermore, the choice of the opioid (H(2) = 2.175, *p* = 0.34, ε^2^ = 0.044) and the severity of extraction (H(7) = 6.282, *p* = 0.51, ε^2^ = 0.128) had no influence on the sedation quality as indicated by a Sheirer–Ray–Hare test.

### 3.4. Pain Scores

A comparison of the median and the maximum of the total pain scores of the EQUUS-FAP, the SUPP and the FAPplusSUPP showed that the opioids had different effects on the pain state (Figure 2). 

Regarding to the EQUUS-FAP, the choice of opioid had a significant influence on the median score (H(2) = 9.31, *p* = 0.02, ε^2^ = 0.19). The median pain score of the group LEV was significantly lower compared with the group BUT (*p* = 0.008).

In the SUPP, the choice of opioid also had a significant influence on the median score (H(2) = 10.61, *p* = 0.004, ε^2^ = 0.22), whereas the LEV group was rated significantly lower than the BUT group (*p* = 0.02). The choice of opioid also had a significant effect on the maximum of the rated score in the SUPP (H(2) = 11.76, *p* = 0.003, ε^2^ = 0.24). Thus, lower maximum total scores were assigned in the LEV group (*p* = 0.005) and in the BUP group (*p* = 0.02) as compared with BUT.

In FAPplusSUPP, the median of the score values was dependent on the choice of opioid (H(2) = 13.33, *p* = 0.007, ε^2^ = 0.27). Both, the LEV group (*p* = 0.001) and the BUP group (*p* = 0.028) had significantly lower median total scores than the BUT group. The choice of the opioid influenced the maximum values of the total score (H(2) = 10.10, *p* = 0.006, ε^2^ = 0.21). As in the EQUUS-FAP and in the SUPP, the LEV (*p* = 0.007) and BUP (*p* = 0.03) groups also achieved in this score lower maximum scores as compared with BUT. LEV and BUP did not differ in their median (*p* = 0.25) or in their maximum values (*p* = 0.48) in the total score.

### 3.5. Development of Pain Level over Time

To compare the development of pain within the first 24 h postoperatively, a generalized additive model on the ordinal scale was used to compare the pain scores over time. Results show that the choice of opioid (namely, LEV and BUP) influenced the development of the severity of pain over time by showing significantly lower pain score values when compared with BUT (Figure 3). Specifically, for EQUUS-FAP: LEV (*p* = 0.006) and BUP (*p* = 0.03); for SUPP: LEV (*p* = 0.0009) BUP (*p* = 0.03); and for FAPplusSUPP: LEV (*p* = 0.0002) and BUP (*p* = 0.009).

### 3.6. Serum Cortisol

The serum cortisol concentration showed significantly higher postoperative values for the opioids LEV (*p* = 0.006) and BUP (*p* = 0.02) compared with BUT (Figure 4). 

### 3.7. Locomotor Activity

The estimation of locomotor activity with pedometers induced by the opioids varied significantly between the groups, as indicated by Kruskal–Wallis test (H(2) = 25.84, *p* < 0.0001, ε^2^ = 0.53) (Figure 5). Significantly more steps were counted in group BUP than in LEV (*p* < 0.0001) and BUT (*p* = 0.00024) 12 h postoperatively compared with the median of the baseline (371 front limb; 357 hind limb).

## 4. Discussion

Our hypothesis was that LEV or BUP are better suited for analgesia protocols for treating somatic pain that is induced by dental diseases and extractions when compared with the commonly used standard protocol that includes BUT as the analgesic. This is due to different properties, such as duration of action and specific receptor affinities [1]. Based on the results of this study, the hypothesis can be evaluated as follows: the use of LEV and BUP did not achieve deeper sedation, did not change the amount of α-2-adrenergic agonist that was required, and did not significantly alter the quality of sedation when compared with the use of the standard protocol with BUT. However, LEV significantly induced less ataxia. Additionally, the pain grading with the pain scores applied in this study showed that in the postoperative phase, both LEV and BUP treatments were able to improve analgesia significantly. 

### 4.1. Sedation and Ataxia

The mean drip rate of the detomidine CRI did not differ between the three groups. None of the opioids therefore appeared to have a greater influence on detomidine-mediated depth of sedation. Although the drip rates did not differ significantly, when scoring ataxia, it was noticeable that the horses in the BUT group were more ataxic than the horses in the BUP and LEV groups (Figure 1). BUT does not seem to affect the depth of sedation but causes a more pronounced muscle relaxation than BUP and LEV at the selected dosage. Potentiation of ataxia without reinforcing the depth of sedation by the addition of BUT was previously reported [35]. Krulijc et al. (2006) reported similar results by using electroencephalographic and electromyographic examination during detomidine-mediated sedation with the addition of BUT [36]. Another study also observed an increased ataxia during detomidine-mediated sedation after the addition of 5 µg kg^−1^ IV BUP [37]. LEV has a modest ataxia-enhancing effect during sedation mediated by α-2-adrenergic agonists [38].

We could not estimate the direct effect that the three opioids had on the extent of ataxia because of the administration of detomidine. However, since the mean total detomidine dose and the additive boli did not differ significantly in any of the groups, it can be concluded that BUT leads to more ataxia when compared with the BUP and LEV treatments.

Another limitation of studying the influence of different effects on sedation quality and postoperative analgesia is the lack of information on equipotent dosages of the respective opioids. The choice of dosages used was made based on personal experience and results of previous studies [6,39,40].

### 4.2. Pain Scores

The results of all three pain scores showed that the increase in pain behaviour caused by the surgery was lower with both LEV and BUP when compared with BUT, while no significant difference was found between BUP and LEV. The proportion of horses with no detectable pain (i.e., pain score = 0) was highest in the LEV group over the entire time course.

The three opioids exhibited different drug–receptor interaction. This can explain the better postoperative analgesia of LEV, which is a synthetic opioid with a pure agonistic effect on the μ-receptor. In comparison, BUT has a high affinity to the κ-receptor but only a moderate effect at the µ-receptor due to agonistic and antagonistic effects [1,41]. Additionally, LEV has a non-competitive antagonistic activity at NMDA receptors that play a role in the development of chronic pain and are part of the development of the wind-up phenomenon, central sensitization and hyperalgesia [21,42]. A study in rats showed that NMDA receptor antagonists are able to prevent the development and maintenance of central sensitization and lead to a significant reduction in hyperalgesia caused by the wind-up phenomenon [43]. Preventing the initial neural cascade with initiation of optimal analgesia could lead to long-term benefits by eliminating the hypersensitivity produced by exnoxious stimuli. Studies demonstrated the benefits of preventing central sensitization such as limitation of subsequent pain experiences [44,45].

BUP is a κ- and δ-receptor antagonist and a partial µ-receptor agonist. It is characterized by fast binding ability and high lipophilia. Due to the high affinity of BUP to the receptor and the associated long-lasting effects, studies in horses showed a significant increase in thermal threshold values of up to seven and nine hours after intravenous administration. It was also shown in thermal and mechanical threshold tests that the analgesic effect of BUP lasted longer when compared with BUT [16,46]; this might also reflect the better analgesia of clinical pain with BUP in this study.

A limiting factor in dental pain evaluation is that a dedicated and validated pain scoring method does not exist, and the fact that the supplemented score that was additionally used in this study is not validated needs to be considered when interpreting our results.

We compared between pain scoring using the EQUUS-FAP and SUPP to evaluate their suitability for the assessment of postoperative dental pain. The score values in the validated EQUUS-FAP rise steadily for all three opioids up to 3 h postoperatively. Beyond this time point, the scores decrease again (Figure 3). However, score values in the unvalidated SUPP presented a steady decrease for both the LEV and BUP groups already from the first examination postoperatively. For the BUT group, time-dependent score values of the unvalidated SUPP were similar to those of the EQUUS-FAP—i.e., an increase of up to 3 h postoperatively followed by a decrease. However, the curve of decreasing score values was not as steep as the scores generated with the EQUUS-FAP (Figure 3). In agreement with previous studies, we concluded that although EQUUS-FAP disregards many dental-specific pain parameters, it is of practical use, as opioid-induced differences on postoperative pain levels were noticeable in this study and due to the lack of a validated alternative [4]. Nevertheless, when comparing between median scores of the EQUUS-FAP (0–18) (Figure 2) and of the SUPP (0–8), it was noticeable that EQUUS-FAP scores were lower over the entire study period. The low suitability of some parameters of the EQUUS-FAP for the assessment of tooth-associated pain was also discussed in previous studies [6]. Several parameters of the EQUUS-FAP such as flehmen or yawning, teeth grinding or moaning rarely changed—neither in the baseline measurements nor in the postoperative period—although the total pain scores were reproducibly increased. 

Furthermore, in the present study, pain-related opened/tightened eyelids was not observed in any patient. Therefore, assessment of this behaviour as well does not seem to be a suitable dental pain evaluation parameter. Sporadically, horses rather showed half-closed eyes instead of wide-open eyes, typically related to pain. 

Inclusion of the additional parameters from SUPP in the FAPplusSUPP adds relevant parameters and helps to clarify the pain curves, facilitating a more sensitive graduation of tooth-associated pain. Nevertheless, SUPP may also require some adjustments. This was evident in this study by looking at the patients’ heart rate, which was not elevated due to dental pain.

Validation of the modified FAPplusSUPP, which includes only parameters that are relevant to the assessment of dental pain, would be beneficial and indeed necessary in order to make reliable assessments.

In a recent study, a new pain score for assessment of chronic pain was successfully applied to horses with dental pain caused by equine odontoclastic tooth resorption and hypercementosis (EOTRH) [47]. This score includes parameters that evaluate feed intake and could be useful for assessing chronic dental pain, such as that expected in EOTRH patients. This pain score is not validated as suitable for pain evaluation following cheek tooth extraction. Pain caused by EOTRH is different in character, and its assessment is not comparable with pain caused by cheek tooth extractions. Lastly, evaluation of the interaction in the herd or the feed intake of apples or carrots make it unsuitable for pain evaluation in the immediate postoperative phase in the clinic. 

### 4.3. Cortisol 

Elevated serum cortisol concentrations could be a result of reaction of the hypothalamus–pituitary axis to an increased pain state of the horses caused by a lower analgesic potency of LEV and BUP. However, this assumption is contradicted by the fact that the pain scores at 180 and 360 min postoperatively were lower in both the LEV and BUP group compared with the BUT group. Both EQUUS-FAP and the SUPP showed that the pain caused by the surgery appeared to be lowest in the LEV group. Another study showed that the application of the racemate methadone has an influence on cortisol secretion without pain as the causal factor. However with the addition of acepromazine, no increased cortisol levels were measured [48]. In our study, the causality between the use of opioids and cortisol level could not be determined. Therefore, the increase in cortisol levels in the LEV group cannot be conclusively substantiated. Elevated cortisol levels, as a stress hormone, can have multifactorial causes. This limits its suitability as a specific pain indicator. 

### 4.4. Locomotor Activity

A possible inhibition of opioid-induced locomotor activity by addition of acepromazine could not be demonstrated in this study. In the study of Haunhorst et al. (2022), horses medicated with BUP showed, in addition to extremely high step numbers, manic behaviour such as tense facial features and hay-washing [6]. These behavioural changes could not be observed in the present study, possibly due to lower chosen dosage (5 µg kg^−1^) compared with the Haunhorst’s study (7.5 µg kg^−1^) [6]. Determining the cause for the reduced manic compulsive behaviour (lower BUP dose or addition of acepromazine) was beyond the scope of this study.

Although the manic behaviour of the horses in this study was less than that described in Haunhorst’s study, the constant movement of the horses made it difficult to observe the horses and thus also the pain scoring.

## 5. Conclusions

The main goal of this study was to compare intraoperative sedation quality and postoperative analgesia when using LEV or BUP as the analgesic opioid as opposed to the commonly used BUT. We showed that while LEV and BUP mediated a sedation with less ataxic horses, the quality of sedation was not rated better or worse with any of the opioids. Based on the results of the applied pain scores, postoperative analgesia was most reliable with LEV, which also led to less ataxia than BUT and BUP. Finally, BUP was also associated with increased locomotor activity. As a conclusion, our results suggest that LEV should be the preferred opioid for cheek teeth extraction in horses. The choice of sedation and analgesic protocol should correspond to the type of pain as well as to the surgical procedure.

## Figures and Tables

**Figure 1 vetsci-09-00174-f001:**
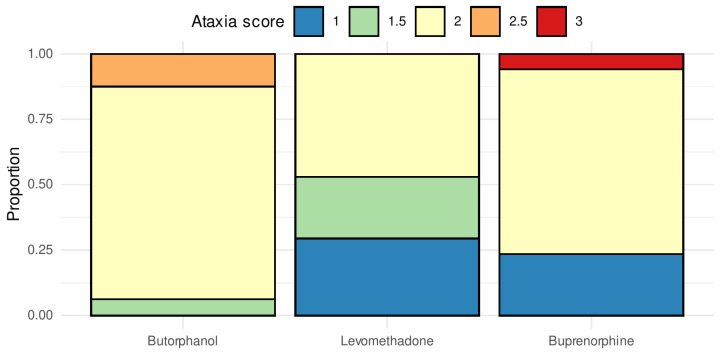
Score of ataxia. Comparison of ataxia according to the used sedation score with a bar plot. Shown is the median of the assigned score values for each opioid on the *X*-axis. The *Y*-axis marks the proportion of horses out of the total number of patients who rated the corresponding median score (1 = mild ataxia; 3 = severe ataxia).

**Figure 2 vetsci-09-00174-f002:**
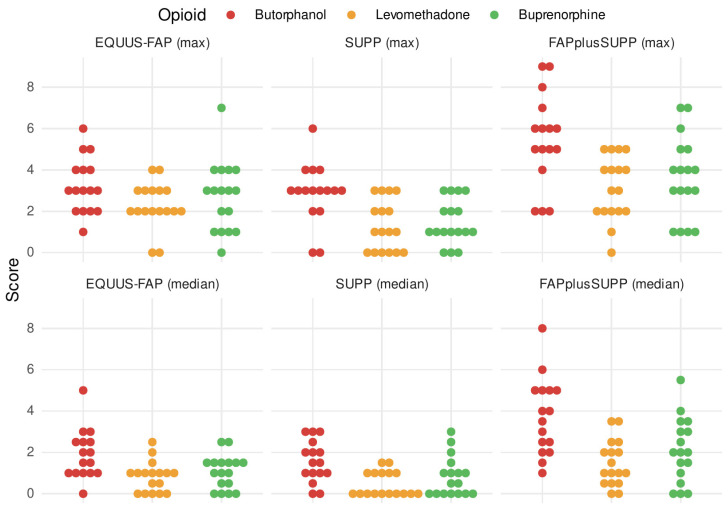
Median and maximum pain scores. Comparison of the total score awarded (maximum = upper row and median = lower row) of the respective pain scores. (From left to right: EQUUS-FAP, SUPP, FAPplusSUPP.) Representation with dot plot.

**Figure 3 vetsci-09-00174-f003:**
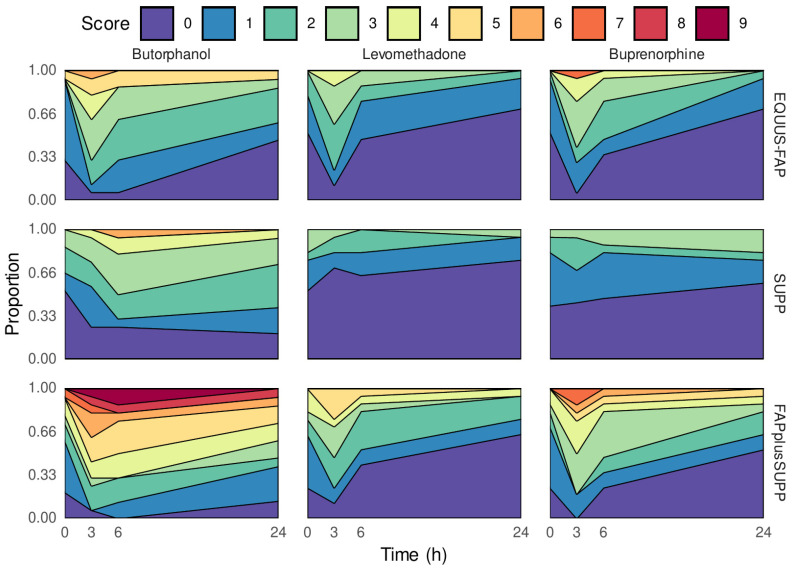
Evaluation of pain development over time using (from top to bottom) EQUUS-FAP, SUPP and the FAPplusSUPP. Total pain score at the respective time of measurement (0–24 h) postoperatively on the *X*-axis and the proportion of horses on the *Y*-axis with the total score awarded.

**Figure 4 vetsci-09-00174-f004:**
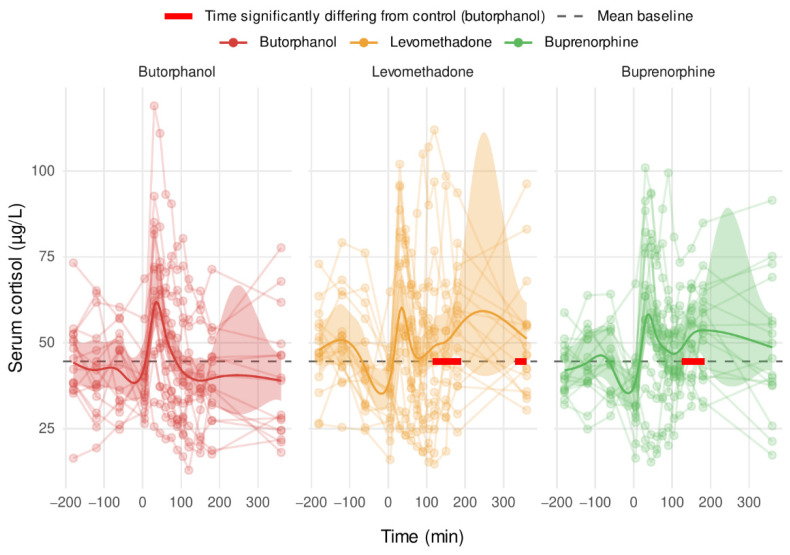
Serum cortisol concentrations of the groups BUT, BUP and LEV over time. The solid line represents the smoothed time course as obtained using a generalized linear model. Its 95% confidence interval is shown as a shaded area in the background. For BUP and LEV, the time windows significantly differing from the control group BUT are represented as red rectangles. Finally, the mean baseline is represented as a dashed black line.

**Figure 5 vetsci-09-00174-f005:**
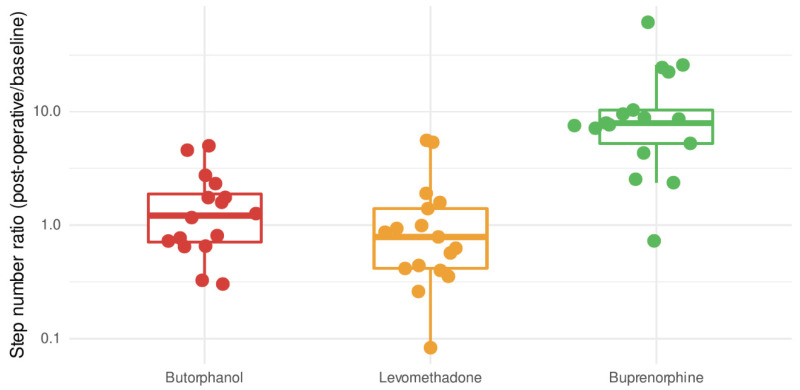
Step number of the treatments (BUT in red, LEV in yellow, BUP in green) normalized by the baseline values (step counts over 12 h without treatment) represented with a Tukey boxplot. Individual data are shown as points. The box and whiskers represent the interquartile range (IQR) and 1.5 IQR respectively. The thick horizontal line is the median.

**Table 1 vetsci-09-00174-t001:** Supplemented portion of the pain score (SUPP); range (0–8).

Score	0	1	2
Palpation of the diseased area	No reaction	Mild reaction	Refused palpation
Heart rate (bpm)	24–44	45–52	>52
Food intake	Present	-	Not present
Difficulties with mastication/ quidding	no difficulties/ no quidding	-	difficulties/quidding observed

**Table 2 vetsci-09-00174-t002:** Demographics of the patients within the groups.

	BUT	LEV	BUP
Total number	16	17	17
Sex (M/F/MC *)	0/7/9	0/9/8	1/7/9
Age (y) Median [Min–Max],	15 [10–21]	13 [12–18]	12 [9–18]
Breed (DH/P/TB/WB **)	1/4/2/9	0/5/2/10	3/1/2/11
Weight (kg) Median [Min–Max],	555 [436–577]	566 [453–539]	537 [430–599]
Jaw [Mxl/Mxl + Mand/Mand ***]	9/1/6	9/0/8	9/0/8

* M = male; F = female; MC = male castrated ** DH = draft horse; P = pony; TB = thoroughbred; WB = warmblood *** Mxl = maxillary; Mand = mandibular.

## Data Availability

Data and materials are available from the corresponding author on reasonable request.

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
