# Peer review of "Influence of Butorphanol, Buprenorphine and Levomethadone on Sedation Quality and Postoperative Analgesia in Horses Undergoing Cheek Tooth Extraction"

_vetsci, 2022, doi:10.3390/vetsci9040174_

Round 1

Reviewer 1 Report

Dear authors, 

Thank you very much for your manuscript. It has been a very interesting reading!

It is very refreshing to see more people using R for treatment of the data and to see such colourful figures!

I have only a couple of remarks/questions:

1) Could you include details of your statistics?

2) On line 64, would it be better to use this citation: Am J Vet Res 1994, 55 (10): 1428-33. Pharmacokinetics and pharmacodynamics of acepromazine in horses. P J Marroum, A I Webb, G Aeschbacher, S H Curry. It seems a better fit than the other ones you have cited. 

3) Did you do a sample size calculation? How did you determine the number of animals you needed to include in your study? 

4) The legends of your figures should be reviewed to make them more understandable 

5) On the section results, you should be explaining what you are reporting. Like (H(2) = 1.21, p = 0.55, ε² = 0.02), with no explanation at any point. 

Apart from that, I am very happy with the manuscript. Material and methods are well described, discussion is well organised and complete (including limiting factors) and conclusions are based on your results. 

Author Response

      We would like to thank you for carefully reading our submitted manuscript and for raising important and insightful remarks/questions that helped us to improve the quality and readability of the manuscript. In what follow below, the comments made by the reviewer are in black and our answers are in red. All changes to the manuscript are marked by “track changes” in the manuscript file.

  1. Could you include details of your statistics?
  • As we don’t know which details you specificaly mean, may you precise your question about the statistical details that are missing?
  • In line 180-213 you can find a detailed explanation of the statistical tests used in this study.
  • Furthermore, I added in the result section now the used tests where it might haven’t been clear
  1. On line 64, would it be better to use this citation: Am J Vet Res 1994, 55 (10): 1428-33. Pharmacokinetics and pharmacodynamics of acepromazine in horses. P J Marroum, A I Webb, G Aeschbacher, S H Curry. It seems a better fit than the other ones you have cited.

- Citation added

  1. Did you do a sample size calculation? How did you determine the number of animals you needed to include in your study?
    - added a section that should answer this question in line 182-187

  1. The legends of your figures should be reviewed to make them more understandable 
    - Thanks a lot for this note! Since there was a mistake in the prior figures (buprenorphine and levomethadone mixed up) that is corrected now, I hope it will be now better to understand

  1. On the section results, you should be explaining what you are reporting. Like (H(2) = 1.21, p = 0.55, ε² = 0.02), with no explanation at any point. 

    - the information given here corresponds to the general recommendations to report not only the P-value but also the test statistic and effect size for a result, this has been implemented wherever possible.
    - furthermore I explained now in l. 201,202 the symbols the Greek symbols for a better understanding

Reviewer 2 Report

The objective of this study was to compare the quality of intraoperative sedation and postoperative analgesia when using LEV or BUP as an analgesic opioid versus commonly used BUT. the study suggests that LEV should be the preferred opioid for cheek tooth extraction in horses. the study reflects the importance of the choice of the anesthetic protocol adequate to the surgical needs in order to have a successful intervention.

Author Response

      We would like to thank you for carefully reading our submitted manuscript and for raising important and insightful remarks/questions that helped us to improve the quality and readability of the manuscript. In what follow below, the comments made by the reviewer are in black and our answers are in red. All changes to the manuscript are marked by “track changes” in the manuscript file.

  1. Improve Conclusion: The objective of this study was to compare the quality of intraoperative sedation and postoperative analgesia when using LEV or BUP as an analgesic opioid versus commonly used BUT. the study suggests that LEV should be the preferred opioid for cheek tooth extraction in horses. the study reflects the importance of the choice of the anesthetic protocol adequate to the surgical needs to have a successful intervention
  • We included the suggested information in the conclusion of our paper

Reviewer 3 Report

Dear authors,

Introduction

In the introduction, it would be advisable not to refer to the commercial name of the pharmacological active principles on the German market, as the article is intended to be universally read. The reference can be made discreetly in the footer.

I suggest a change in the order of description of drugs in the introduction, according with their appearance order in title and results.

Lines 52 - NMDA (N-methyl-D-aspartate) receptor antagonists

Material and Methods

 Lower saxony state office for consumer protection and food safety (LAVES)

Lines 143 to 145 -  you should provide this scale:

"the quality of sedation, surgical conditions, and the severity of the extraction procedure was assessed by the surgeon using a numerical rating scale (NRS)" 

Results

statistical "p" in italics

Figures should be previously announced in the text as it happens with Figure 2

Lines 204 to 206 - I cannot find in the material and methods section a reference to a comparative study between groups regarding the detomidine additives in the bolus form nor on the average drip rate of the CRI between the groups

References

Line 529 - Thomson, P J; Rood JP. Preemptive analgesia reduces postoperative pain experience following oral day case surgery. Ambul Surg 1995;3.

Author Response

      We would like to thank you for carefully reading our submitted manuscript and for raising important and insightful remarks/questions that helped us to improve the quality and readability of the manuscript. In what follow below, the comments made by the reviewer are in black and our answers are in red. All changes to the manuscript are marked by “track changes” in the manuscript file.

Introduction

  1. In the introduction, it would be advisable not to refer to the commercial name of the pharmacological active principles on the German market, as the article is intended to be universally read. The reference can be made discreetly in the footer.
  • Changed it as suggested in the paper

  1. I suggest a change in the order of description of drugs in the introduction, according with their appearance order in title and results.
  • Order changed as suggested

  1. Lines 52 - NMDA (N-methyl-D-aspartate) receptor antagonists
  • Abbreviation NMDA explained as suggested (l.59 in this new version of the manuscript)

Material and Methods

  1. Lower saxony state office for consumer protection and food safety (LAVES)
  • Suggestion followed and changed

  1. Lines 143 to 145 - you should provide this scale: "the quality of sedation, surgical conditions, and the severity of the extraction procedure was assessed by the surgeon using a numerical rating scale (NRS)" 
  • We provided NRS scale in the text: line 151-155. If you are you referring to the sedation score, we provided this one now as well (as Supplemental S1)

Results

  1. statistical "p" in italics
  • Changed as suggested

  1. Figures should be previously announced in the text as it happens with Figure 2
  • Adapted to all the Figures

  • Lines 204 to 206 - I cannot find in the material and methods section a reference to a comparative study between groups regarding the detomidine additives in the bolus form nor on the average drip rate of the CRI between the groups

  • Here I am not sure if I understood this question correct
  • There was no comparison made / we didn’t distinguish between the weight/breed age and sex and their detomidine bolus or average drip rate
  • I added a reference in the Material and Methods section with a study on which base we decided for the dosages of the boli and the detomidine dripping rate (l. 139)

References

  1. Line 529 - Thomson, P J; Rood JP. Preemptive analgesia reduces postoperative pain experience following oral day case surgery. Ambul Surg 1995;3.
  • Corrected